# Scenario-Mining for Level 4 Automated Vehicle Safety Assessment from Real Accident Situations in Urban Areas Using a Natural Language Process

**DOI:** 10.3390/s21206929

**Published:** 2021-10-19

**Authors:** Sangmin Park, Sungho Park, Harim Jeong, Ilsoo Yun, Jaehyun (Jason) So

**Affiliations:** Department of Transportation Engineering, Ajou University, Suwon 16499, Korea; stylecap@ajou.ac.kr (S.P.); fenix3339@ajou.ac.kr (S.P.); gkfla0731@ajou.ac.kr (H.J.); ilsooyun@ajou.ac.kr (I.Y.)

**Keywords:** automated vehicle, scenario-mining, safety, natural language process, accident data

## Abstract

As the research and development activities of automated vehicles have been active in recent years, developing test scenarios and methods has become necessary to evaluate and ensure their safety. Based on the current context, this study developed an automated vehicle test scenario derivation methodology using traffic accident data and a natural language processing technique. The natural language processing technique-based test scenario mining methodology generated 16 functional test scenarios for urban arterials and 38 scenarios for intersections in urban areas. The proposed methodology was validated by determining the number of traffic accident records that can be explained by the resulting test scenarios. That is, the resulting test scenarios are valid and represent a matching rate between the test scenarios and the increased number of traffic accident records. The resulting functional scenarios generated by the proposed methodology account for 43.69% and 27.63% of the actual traffic accidents for urban arterial and intersection scenarios, respectively.

## 1. Introduction

The test scenario is a key measure for evaluating and ensuring the driving capability of automated vehicles (AVs) [1]. For validity and effectiveness, the test scenarios should be concretized with the road geometry [2,3,4,5], traffic situations, and microscopic vehicle maneuvers in detail, and should represent complicated road traffic conditions as well as dangerous vehicle maneuvers. Among the many types of road sections, urban roads are known to be the most complicated because of various traffic controls, many entry/exit points, and a variety of road users [6]; such conditions are a threat to AVs and degrade the performance of automated driving. There are various dangerous traffic situations on urban roads and AVs should be tested under these conditions to ensure that they can be used on the roads.

However, it is challenging to generate representative test scenarios because there are many traffic situations on urban roads [7]. Various data sources such as traffic cameras and AVs sensor [8] datasets can be used to derive various situations on urban roads; however, such datasets are too extensive to analyze [9,10] and extracting only unsafe traffic conditions is more challenging [11]. Traffic accident data [12] are useful for extracting unsafe traffic conditions, critical objects, and behaviors that cause accidents. In particular, traffic accident data include details of the road types and geometric characteristics, traffic conditions, and vehicle maneuvers before and during an accident, which can be used to construct test scenarios.

With the advantage of traffic accident data in extracting test scenarios, this study aims to develop a test scenario-mining methodology using a natural language processing technique (NLP). That is, the NLP technique is used to analyze the texts [13] in accident investigation reports and extract important elements (e.g., road types and geometric characteristics, traffic conditions, and vehicle maneuvers) that cause an accident. It should be noted that the traffic accident report includes both structured data consisting of code numbers and text-based data written by police officers. 

To carry out this study, Section 2 reviews previous studies on automated vehicle test scenarios and application of NLP technique in traffic engineering field. Section 3 presents the premise of the automated vehicle driving capability and operational design domain (ODD) [14] used in this study. Section 4 presents a test scenario-mining methodology for generating automated vehicle safety assessment scenarios from traffic accident data. Section 5 presents the results of the derived features and functional scenarios mined by both the proposed methodology and traffic accident data collected from urban arterial roads in Korea. Also, to verify the derived functional scenarios, a verification process is conducted Section 6 presents the conclusions. Section 7 presents future research tasks.

## 2. Literature Review

To develop safety assessment scenarios for AVs, prior research related to the safety assessment of AVs was reviewed in this study. Choi and Lim [15] developed a scenario for the AEBS test using National Highway Traffic Safety Administration (NHTSA) traffic accident data and an automotive collision case catalog, which is a type of Korean traffic accident data source. A PC-Crash simulation was used for the data analysis. Additionally, traffic accident data were analyzed to develop scenarios according to the road types and traffic accident types. From the analysis, the road types where traffic accidents occurred in Korea were divided into five types and the collision types were divided into six types. Moreover, based on the analysis results, ten accidents at intersections and five accidents on road sections were classified. Finally, a total of 3960 AEBS scenarios were developed using the velocity, angle, and offset at the collision. Zhu et al. [16] proposed a method that uses optimization searching to generate parameters for automated vehicle risk scenarios. The proposed method included five modules. The five modules are composed of an exploration and exploitation module, moving probability determination, step size determination, a memory function module, and a result analysis module. The proposed method could quickly find the risk parameter space in a given logical scenario. Nalic et al. [17] developed a co-simulation framework to develop scenarios for the evaluation and verification of AVs. A method combining the IPG CarMaker and PTV VISSIM was proposed for co-simulation development. In the proposed method, data were processed for every simulation cycle and a new scenario was constructed. All tests that ran data were saved with the relevant scenarios, confirming that real traffic scenarios can be created indefinitely and used for testing. Holland and Sargolzaei [18] proposed a methodology to create and verify automated vehicle scenarios based on actual automated vehicle traffic accidents. The proposed method makes the design of a virtual road possible using the accident location of an automated vehicle and map data based on the accident location. Moreover, an accident scenario was generated using automated vehicle accident description data using a natural language processing (NLP) technique. The generated scenario included the actual characteristics of the road, such as the curvature and number of lanes. So et al. [19] presented a methodology for developing automated vehicle test scenarios. Based on the big data technique, the descriptions of the crash data were analyzed with a C# -based automated analysis program and an automated vehicle test scenario was verified with a combination of 19 frequently mentioned keywords. As a result, scenarios were derived from road and intersection sections. As a target object, this research study developed only vehicle-to-vehicle scenarios. Menzel et al. [20] presented the terminology and experimental process of an automated vehicle experiment scenario. The concept and experimental stages of functional, logical, and concrete scenarios were defined according to the level of abstraction of the scenario contents. Waymo [21] was used to test a total of 47 functions by adding 19 behavioral functions to the 28 behavioral functions that AVs should perform, as recommended by the NHTSA. A scenario was derived and tested using an automated vehicle developed by Waymo. Additionally, to test the collision avoidance capability, based on the NHTSA’s pre-collision scenarios, 28 scenarios in four categories were derived and the scenarios were added based on the experimental results in the future. Chen and Kloul [22] proposed an approach to automatically generate use cases of AVs for highways. The proposed approach combined three ontologies as a knowledge base for the generation of highway scenarios, including highway, weather, and vehicle. Relationships and rules, such as traffic regulation, were expressed using first order logic. Ghodsi et al. [23] presented a method of generating and characterizing scenarios for AV safety testing. In the study, the authors used next generation simulation (NGSIM) data to characterize the real driving and adversarial scenarios generated in the simulation. The proposed methodology could generate 240 unsafe scenarios per hour. Riedmaier et al. [24] surveyed a scenario-based safety assessment of AVs. In this research study, there were two ways of scenario generation and extraction that were knowledge-based and data-driven. According to the authors, an infinite number of different scenarios can theoretically occur in real-world traffic, thus finding representative scenarios is important.

Additionally, the present study reviewed research related to NLP technique, which is a method in which computers can understand documents or words written by human languages [25]. NLP is used for traffic engineering area to extract important information for the analysis of traffic accidents and traffic management. 

Kamerkar et al. [26] used a text-mining technique to analyze rail accidents in India. This study proposed ensemble methodology using text data and the proposed methodology could automatically discover characteristics of rail accidents. Cheng et al. [27] used a machine learning technique based on NLP to classify accidents in construction site. This study used accident reports to extract relevant knowledge and information, which can be valuable to prevent future accidents in construction projects. Gao et al. [28] developed a verb-based NLP technique using traffic accident data. This study collected and used traffic accident data from the Missouri State Highway Patrol from 19 May to 27 June 2012. As a result of the analysis, it was shown that important information can be extracted for understanding traffic accidents. Ali et al. [29] developed fuzzy ontology and LSTM-based text-mining methodology to monitor the transportation network for assisting travel. This study used an NLP technique to extract relevant features from user-generated text on social media. This study showed that text data generated by social media users can be used for transportation entities or feature extraction. Ali et al. [30] proposed a traffic accident detection and condition analysis framework using social networking data and NLP techniques, such as OLDA, word embedding, and Bi-LSTM. This study showed that the proposed framework can offer detection and analysis that extracts the most valuable traffic information from unstructured data and represents accurately detected and analyzed traffic situations.

## 3. Premise of Scenario Development

### 3.1. Automated Vehicle Driving Capability

To develop safety assessment scenarios, it is necessary to define the automated vehicle capabilities in advance, such as the automated vehicle level and functions. To this end, the technology level, which is the Society Automotive Engineers (SAE) [31] of the target automated vehicle, was defined.

This study selected level 4 for target automated vehicles, which is a high automation that AVs can have independently without providing control to the driver in unsafe traffic conditions. However, because the level 4 fully automated vehicle is currently under development, there is a problem in that it is difficult to determine the exact specifications. Thus, the range of capabilities of the automated vehicle was defined. The behavioral competency of the target automated vehicle was defined as stop and go, lane-change, passing through a signalized intersection, and turning an intersection. This automated vehicle is capable of cooperative driving using V2X communication, which leads to the communication and recognition of traffic signals at signalized intersections. It was assumed that the automated vehicle defined in this study does not cause malfunction and drives itself by following the given driving rules and laws.

### 3.2. Operation Design Domain

For an automated vehicle to drive properly, it is necessary to set the drivable areas and conditions. Defining the drivable areas and conditions is referred to as ODD. Currently, ODD has various definitions in many international standards. According to the ISO 21448 standard [32], ODD is “the specific conditions under which a given driving automation system is designed to function”. Specific conditions include spatial, temporal, and environmental conditions.

In this study, spatial, temporal, and environmental conditions were defined for AVs. The spatial-specific conditions, road type, number of lanes, and speed limit were selected. To define the road type, this research considered continuity with expressways and selected urban arterial roads, including roads and intersections. This is because the urban arterial road is the road where AVs will be introduced, next to the expressway, after the expansion to other roads to reach the destination. Figure 1 shows the concept of extended odd from expressway to urban arterial roads.

To find specific elements, such as the number of lanes and speed limits, a Korean national standard node link GIS map was used. Next, the temporal condition was defined as 7:00–18:00 h, which is in the daytime. Finally, the weather was considered the environmental conditions and defined as sunny. The defined ODD is expressed as shown in Table 1.

## 4. Methodology

### 4.1. Overview

This study proposes a scenario-mining methodology for the automated vehicle safety assessment from traffic accident data using an NLP technique. Considering the traffic accident data includes the ‘accident situation description’ described in the text, it is possible to understand the traffic accident situation. In this study, automated vehicle scenarios were developed by extracting traffic accidents that occurred on arterial roads in urban areas and using the proposed methodology. The proposed methodology consists of six steps: data collection, data extraction, data preprocessing, feature extraction, feature categorization by object, and scenario-mining. Figure 2 shows the structure of the proposed research methodology.

### 4.2. Data Collection

To develop a scenario for the automated vehicle safety assessment, this study utilized general automobile traffic accident data managed by the Korean National Policy Agency (KNPA). To assess the safety of AVs, it is best to use AV traffic accident data. However, the existing AV traffic accident data remains insufficient to generate scenarios. In addition, in mixed traffic conditions with human-based objects such as vehicles, pedestrians, motorcycles, and bicycles, AVs would encounter dangerous situations caused by human-based objects, as general vehicles have encountered as well. Thus, KNPA automobile traffic accident data could be an alternative to develop AV scenarios.

The KNPA traffic accident data include various data such as time, location (GPS coordinates), accident type, vehicle type, and accident situation descriptions. Therefore, it is possible to analyze the object and situation that caused the traffic accident at the time. Particularly, the ‘accident situation description’ describes the situation in the event of a traffic accident, written in text. In this study, 223,552 traffic accident data from 2014 were collected nationwide to perform the scenario-mining.

To extract the traffic accident data that occurred on urban arterial roads in Korea, it was necessary to extract only the relevant traffic accidents from the collected data. As the traffic accident data of the KNPA include GPS coordinates, it is possible to extract the relevant traffic accident by performing spatial join using GIS software. Therefore, traffic accidents that occurred in urban arterial roads and at intersections were extracted through spatial join with the accident data, defined ODD, and GIS map. In this study, spatial join was performed using ArcGIS 10.3, a representative GIS tool. As a result of extracting data through spatial join, 2824 road sections and 4166 intersection sections were extracted.

### 4.3. Data Preprocessing

To utilize text data, which is an “accident situation description” from the accident data, preprocessing is essential. To this end, the Python 3.7 and Mecab library, which is the predominant Korean natural language process library, was utilized for data preprocessing.

In this study, we reviewed and selected a text data preprocessing technique. Ten preprocessing techniques are frequently used [33]. The data preprocessing has four steps: data cleansing, similar word matching, stop word removal, and tokenization. In the data cleansing step, this study removed text such as punctuation marks, special characters, numbers, etc., which cannot grasp the meaning from the data. A similar word matching step could address synonyms because different people may use different words to record accident situation descriptions. A stop word removal step was then performed. Stop words are common words with no semantics and do not aggregate relevant information to the task, such as “the” and “a” [33]. Lastly, the tokenization step divides each accident situation description sentence into token units, which are small chunks such as words and attached parts of speech. In particular, in this study, among several parts of speech, nouns, including compound nouns, were used.

### 4.4. Feature Extraction

To extract features, which are meaningful words, from text data, the feature extraction process is essential. To select the relevant feature extraction method, this study reviewed feature extraction methods. There are four feature extraction methods including bag-of-words, the term frequency-inverse document frequency (TF-IDF) model, and Word2Vec, which is mainly used in the NLP [34].

This study selected the TF-IDF model, which is the most widely used in NLP and has the advantage of expressing the relative importance of each word in an individual document. Additionally, the TF-IDF model is able to provide weight to words that appear frequently throughout the document rather than simply applying weight by the frequency [35]. Equation (1) is a TF-IDF model.
(1)$$TF-IDF_{\left(w,d\right)}=TF_{\left(w,d\right)}\times \log\left(\frac{n}{DF_{\left(w\right)}}\right)$$
where $TF_{\left(w,d\right)}$ = number of words, w, in documents, d; n = number of total documents; and $DF_{\left(w\right)}$ = number of documents including words, w.

Using the TF-IDF model, this study derived features and TF-IDF values from the collected data of the urban arterial roads and intersections. To derive more meaningful features, trivial features such as the area names, proper nouns, and vehicle names were removed. After that, features were categorized into target objects, maneuvers, provoking events, and so on to determine the meaning of the features.

### 4.5. Feature Categorization by Objects

Each of the derived features has its own meaning but there is a limitation in explaining the corresponding accident situation that contains the features. However, although features also occur individually, they tend to occur together in a specific object. For example, in a traffic accident situation related to vehicle-to-vehicle accidents, a collision due to a stop may occur, but crosswalk-crossing does not occur. Therefore, feature categorization was performed by objects such as vehicles, pedestrians, bicycles, and motorcycles. The features by object were categorized by the accident location, maneuver, target object, and provoking event.

### 4.6. Generation of Functional Scenarios

To develop scenarios for the automated vehicle safety assessment, this study utilized the functional scenario concept proposed by the Pegasus project. This is a project for the establishment of generally accepted quality criteria, tools, and methods, as well as scenarios and situations, for the release of highly automated driving functions, organized under the initiative of the German Federal Ministry for Economic Affairs and Energy [36,37]. A functional scenario is one in which road sections, fixed and dynamic elements, and situations are described in natural language with a high level of abstraction [38].

To develop functional scenarios, this study used previously derived feature categories, maneuvers, target objects, and provoking events, and developed a scenario development system. Specifically, in an accident situation, the object causing the accident was defined as the target object. An action that caused an accident was defined as a provoking event. The victim vehicle was defined as an ego-vehicle and the driving situation at the time was defined as the maneuver of the ego-vehicle. For example, in the situation ‘Vehicle 1 which was driving in the opposite direction hit vehicle 2 which was driving straight’, the features are extracted such that ‘vehicle 1’ is the target object, ‘driving in the opposite direction’ is the provoking event of the target object, ‘vehicle 2’ is the victim vehicle defined as the ego-vehicle, and ‘driving straight’ is the maneuver of the ego-vehicle. Finally, the features were constructed into a functional scenario. Figure 3 depicts the procedure in which the accident data were composed into a functional scenario. It should also be noted that this study includes only the traffic interactions and situations in which AVs are spontaneously involved, while the situations in which normal vehicles crash into the back of AVs, which is unavoidable from the view of AVs, are excluded.

## 5. Results

### 5.1. Key Feature Extraction Results

The features of the road sections and intersections were extracted using the Python and TF-IDF model. For the road sections, 2811 features were extracted. However, since there were insignificant features that could not depict accident situations, such as specific location (municipality), building name, subway station name, vehicle’s brand/maker, and so on (e.g., Seoul, apartment, Sonata, Sadang station, etc.), postprocessing was performed to remove such insignificant features. After postprocessing, fifteen main features were obtained and categorized into object, maneuver, and provoking events. Consequently, vehicles, pedestrians, bicycles, and motorcycles were extracted as the objects; in the case of maneuvering, driving straight, and lane-change, crossing, stop, and U-turn were obtained as the provoking events. Table 2 shows obtained features on the road section.

For the intersection sections, 4096 features were extracted and a total of 15 features were obtained by removing the insignificant features. The main features were categorized by object, maneuver, and provoking event. Among the obtained main features, the object items were extracted as vehicles, bicycles, motorcycles, and pedestrians, and in the case of maneuvers, this research extracted driving straight, left-turn, and right-turn. Stopping, lane-change, crossing, and violating traffic signals were obtained as the provoking situations. Table 3 shows obtained features on the intersection section.

### 5.2. Feature Categorization by Objects

To analyze the obtained features in detail, the main features were extracted by classifying them according to the target object. To obtain the corresponding word from the text analysis, meaningful words were obtained simultaneously rather than single words alone. In the case of a vehicle in the road section, lane-change, stopping, U-turn, etc., were obtained. The driving over centerline and reversing were obtained. Crossing, walking, and jaywalking were obtained for the pedestrians. In the case of walking, it means walking on the road and not walking on the pedestrian. For motorcycles, lane changes, crossings, U-turns, and stops were obtained. In the case of bicycles, crossing, reversing, and straight driving were obtained. Table 4 shows obtained features in the road section by objects.

In the case of the intersection sections, the maneuver of the vehicle was obtained as driving straight, left-turn, and right-turn. From the object analysis, in the case of a vehicle, stopping, lane-change, violating traffic signal, U-turn, and abrupt stopping were obtained.

Crossing, walking, and jaywalking were obtained for pedestrians. In the case of motorcycles, lane-change, stop, violating traffic signals, and crossings were obtained. In the case of bicycles, crossing, reversing, stopping, and violating traffic signals were obtained. Table 5 shows obtained features in the intersection sections by objects.

### 5.3. Scenario Development Results

Using the features obtained from the objects and the scenario development system, functional scenarios of road sections and intersections of urban arterial roads were developed. For the road section, a total of 16 scenarios were derived. All derived scenarios for the road section are presented in Appendix A. In the case of a vehicle as a target object, seven scenarios were developed, as presented in Table A1. In the case of a pedestrian as a target object, three scenarios were derived, as presented in Table A2. In the case of a motorcycle as a target object, three scenarios were derived, as presented in Table A3. In the case of a bicycle as a target object, three scenarios were derived, as presented in Table A4. Table 6 shows examples of developed functional scenarios in road sections.

For the intersection sections, a total of 38 scenarios were obtained. All derived scenarios for the intersection sections are presented in Appendix B. In the case of a vehicle as a target object, sixteen scenarios were developed, as presented in in Table A5. In the case of a pedestrian as a target object, three scenarios were developed, as presented in Table A6. In the case of a motorcycle as a target object, sixteen scenarios were developed, as presented in Table A7. In the case of a bicycle as a target object, three scenarios were developed, as presented in Table A8. Table 7 presents an example of developed functional scenarios in intersection sections.

### 5.4. Verification of the Resulting Scenarios

To verify the derived functional scenario, a verification process was performed. This research verified the number of functional scenarios that occur in real traffic accidents in road and intersections. For road sections, the functional scenarios developed in this study accounted for 43.69% of the actual traffic accidents. Vehicle-to-vehicle functional scenarios accounted for 39.35% of the actual traffic accidents in road sections. The ratio of vehicle-to-vehicle functional scenarios from real accident data in road sections is shown in Table 8.

Regarding vehicle-to-pedestrian functional scenarios, they account for 2.10% of the actual traffic accidents in road sections as shown in Table 9.

Regarding vehicle-to-motorcycle functional scenarios, they accounted for 1.07% of the actual traffic accidents in road sections as shown in Table 10.

Regarding vehicle-to-bicycle functional scenarios, they accounted for 1.17% of the actual traffic accidents in road sections as shown in Table 11.

For the intersection sections, the developed functional scenarios in this study were found to account for 27.63% of the actual traffic accidents. Regarding vehicle-to-vehicle functional scenarios, they accounted for 19.8% of the actual traffic accidents at intersection sections. Table 12 shows ratio of vehicle-to-vehicle functional scenarios from real accident data at intersection sections.

Regarding vehicle-to-pedestrian functional scenarios, they accounted for 0.58% of the actual traffic accidents at intersection sections as shown in Table 13.

Regarding vehicle-to-motorcycle functional scenarios, they accounted for 6.70% of the actual traffic accidents at intersection sections as shown in Table 14.

Regarding vehicle-to-bicycle functional scenarios, they accounted for 0.55% of the actual traffic accidents at intersection sections as shown in Table 15.

## 6. Conclusions

As the research and development activities of AVs have been active in recent years, developing test scenarios and methods has become necessary to evaluate and ensure the safety of AVs. Therefore, this study developed an automated vehicle test scenario derivation methodology using traffic accident data and an NLP technique. First, the level of AVs for the scenario to be developed was defined. The level of the automated vehicle was defined as level 4 of the SAE standards, which is high automation, and the ODD was defined as centered on urban arterial roads. Using the ODD defined above, the collected traffic accident data archived by the KNPA were used to extract traffic accidents in road sections and intersections of urban arterial roads. Additionally, the ‘accident situation description’ data described as text among the extracted traffic accident data were preprocessed. The main features were extracted from the preprocessed data using a feature extraction module based on TF-IDF vectorization. Furthermore, the main features of each object were extracted and classified according to the defined categories. 

As a result, 16 functional test scenarios for urban arterials and 38 scenarios for intersections were generated on urban roads. The resulting test scenarios were validated by determining the number of traffic accident records that can be explained by the resulting test scenarios. That is, the resulting test scenarios are valid and represent a matching rate between the test scenarios and the increased number of traffic accident records. The resulting functional scenarios generated by the proposed methodology account for 43.69% and 27.63% of the actual traffic accidents for the urban arterial and intersection scenarios, respectively. Therefore, it is certain that the scenario-mining methodology proposed in this study can derive automated vehicle safety assessment scenarios from traffic accident data and it is inferred that it can be used to develop automated vehicle evaluation scenarios. This proposed methodology can fully utilize traffic accident data that include unsafe traffic conditions and is a systematic method for extracting edge cases, in which AVs need to be tested. Particularly, the methodology provides a practical method to analyze abundant text-based data written by police officers of traffic accident reports, which is barely possible because of the vastness of the data. Finally, this proposed methodology is universal for other traffic accident databases such as the German in-depth accident study (GIDAS), the initiative for the global harmonization of accident data (IGLAD), and the national automotive sampling system crashworthiness data system (NASS CDS), considering such datasets include the data elements used in this study.

## 7. Recommendations for Future Research

Although this study developed a methodology for mining functional scenarios for automated vehicle safety assessment using traffic accident data, NLP techniques, and a scenario for urban arterial roads, some limitations still exist. First, to derive various dangerous situations occurring in road sections, the scenario was derived using the accident situation described in the text of the traffic accident data of the KNPA. Although the traffic accident data represent the accident situation, there is a disadvantage in that detailed information, such as the speed of the vehicle at the time of the accident, the collision angle, and the location of the surrounding vehicles, remain unknown. If CCTV data or individual vehicle sensing data can be used in the future, more detailed scenarios can be derived and configured. Second, there is a limitation in that the methodology cannot be automated to select extracted features and type them by category. It is necessary to categorize the accident situation and derive characteristics using topic modeling or sentence-based embedding in the future. Third, there is a limitation in not evaluating and validating the developed functional scenario as an automated actual-vehicle experiment or simulation experiment. To solve this problem, it is necessary to evaluate and validate the appropriateness of the scenario through simulation or actual-vehicle tests by extending the developed functional scenario to logical and concrete scenarios. Lastly, the methodology needs to be advanced to address the cases in which multiple objects are involved at the same time, as this study focuses only on single object-related accident cases. 

## Figures and Tables

**Figure 1 sensors-21-06929-f001:**
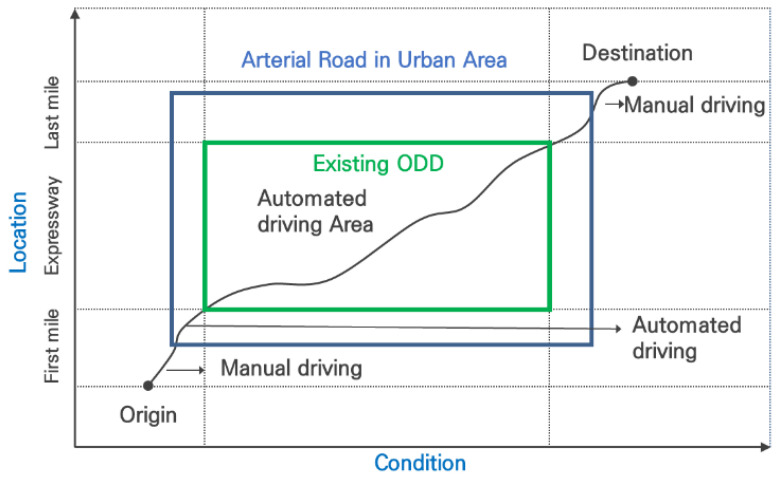
Concept of extended ODD to urban arterial roads.

**Figure 2 sensors-21-06929-f002:**
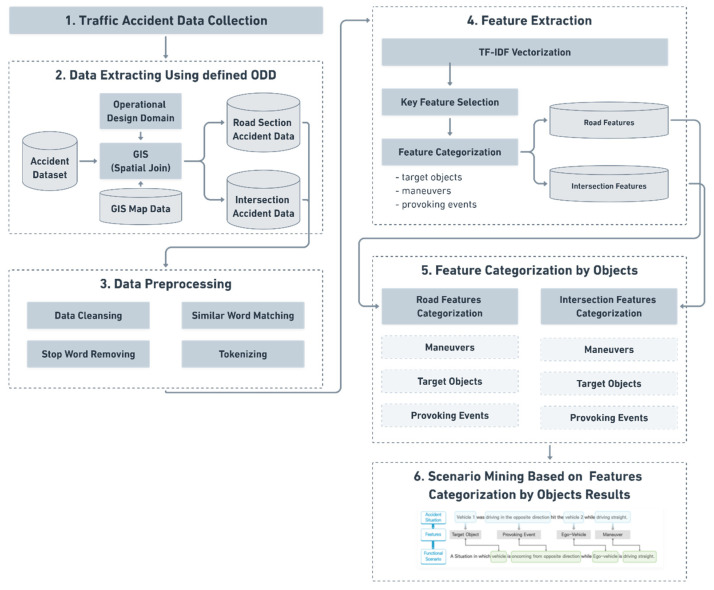
Architecture of scenario-mining process.

**Figure 3 sensors-21-06929-f003:**
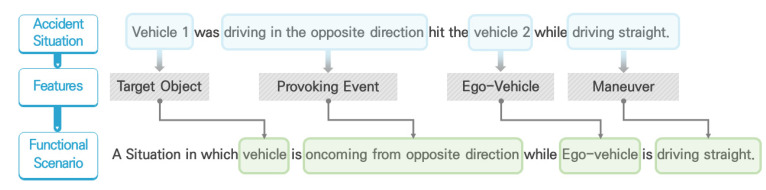
Functional scenario development procedure.

**Table 1 sensors-21-06929-t001:** Defined ODD for urban arterial roads in Korea.

Categories	Items	ODDs
Spatial condition	Road type	Urban arterial road including signalized intersection in Korea
	Number of lanes	4~8
	Speed limit	30~90
Temporal condition	Time	Daytime
Environmental condition	Weather	Sunny

**Table 2 sensors-21-06929-t002:** Obtained features on the road section.

Section	Objects	Maneuver	Provoking Events
Road	Vehicle (1131.54)	Driving Straight (527.30)	Lane-change (254.64)
	Pedestrian (143.36)	-	Crossing (135.75)
	Bicycle (116.34)	-	Stopping (93.19)
	Motorcycle (111.86)	-	U-turn (82.33)
	-	-	Abrupt stopping (44.10)
	-	-	Driving over centerline (37.43)
	-	-	Walking (35.12)
	-	-	Abrupt lane-change (31.52)
	-	-	Jaywalking (19.09)
	-	-	Reversing (7.48)

(): TF-IDF value.

**Table 3 sensors-21-06929-t003:** Obtained features on the intersection section.

Section	Objects	Maneuvers	Provoking Events
Intersection	Vehicle (1583.78)	Driving straight (769.30)	Stopping (291.44)
	Bicycle (190.394)	Left-turn (413.54)	Lane-change (214.32)
	Motorcycle (171.20)	Right-turn (311.90)	Crossing (213.67)
	Pedestrian (166.60)	-	Violating traffic signal (141.34)
	-	-	U-turn (121.97)
	-	-	Walking (43.00)
	-	-	Abrupt stopping (32.98)
	-	-	Jaywalking (14.68)

(): TF-IDF value.

**Table 4 sensors-21-06929-t004:** Obtained features on the road section by objects.

Location	Maneuver	Objects	Provoking Events	
			Features	TF-IDF
Road	Driving Straight	Vehicle	Lane-change	292.67
			Stopping	104.04
			U-turn	78.83
			Abrupt stopping	48.77
			Driving over centerline	36.43
			Abrupt lane-change	34.08
			Reversing	5.27
		Pedestrian	Crossing	51.34
			Walking	18.45
			Jaywalking	11.96
		Motorcycle	Lane-change	23.67
			Crossing	11.54
			U-turn	7.58
			Stopping	4.65
		Bicycle	Crossing	29.25
			Driving straight	12.13
			Reversing	1.36

**Table 5 sensors-21-06929-t005:** Obtained features at the intersection sections by objects.

Location	Maneuvers	Objects	Provoking Events	
			Features	TF-IDF
Intersection	Driving straight	Vehicle	Stopping	396.21
			Lane-change	294.10
	Left-turn			
			Violating traffic signal	169.77
			U-turn	125.03
	Right-turn			
			Abrupt stopping	40.44
	Driving straight	Pedestrian	Crossing	104.26
			Walking	41.03
	Left-turn			
			Stopping	20.91
	Right-turn			
			Jaywalking	15.57
	Driving straight	Motorcycle	Lane-change	26.54
			Stopping	20.45
	Left-turn			
			Violating traffic signal	20.20
	Right-turn			
			Crossing	16.16
	Driving straight	Bicycle	Crossing	19.03
	Left-turn		Stopping	2.52
	Right-turn		Violating traffic signal	1.64

**Table 6 sensors-21-06929-t006:** Examples of developed functional scenarios on road sections.

No.	Depictions	Maneuvers	Provoking Events	Functional Scenarios
1	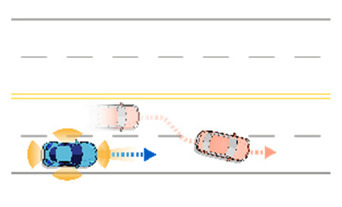	Driving straight	Lane-change	Ego-vehicle is driving straight at road section. Target object (Vehicle) is lane-changing into ego-vehicle’s driving lane ahead.
2	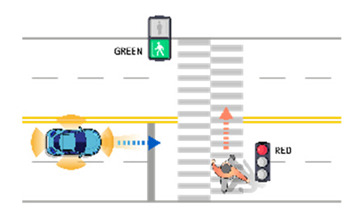	Driving straight	Crossing	Ego-vehicle is driving straight at road section. Target object (Pedestrian) is crossing ahead.
3	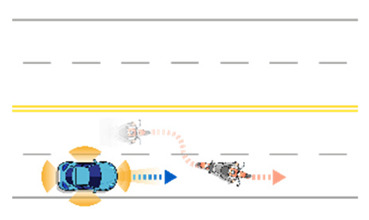	Driving straight	Lane-change	Ego-vehicle is driving straight at road section. Target object (Motorcycle) is lane-changing into ego-vehicle’s driving lane.
4	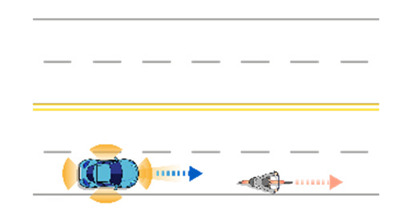	Driving straight	Driving straight	Ego-vehicle is driving straight at road section. Target object (Bicycle) is driving straight into ego-vehicle’s lane ahead.

**Table 7 sensors-21-06929-t007:** Examples of developed functional scenarios at intersection sections.

No.	Depictions	Maneuvers	Provoking Events	Functional Scenarios
1	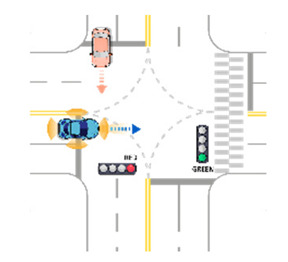	Driving straight	Violating traffic signal	Ego-vehicle is driving straight on the right of way at signalized intersection. Target object (Vehicle) that violates a traffic signal from the left is driving straight.
2	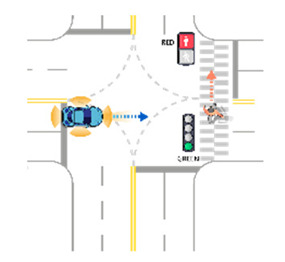	Driving straight	Jaywalking	Ego-vehicle is driving straight on the right of way at signalized intersection. Target object (Pedestrian) is jaywalking ahead at crosswalk.
3	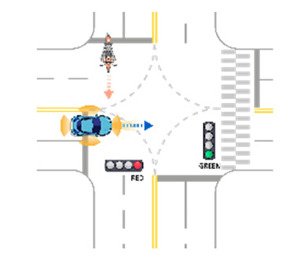	Driving straight	Violating traffic signal	Ego-vehicle is driving straight on the right of way at signalized intersection. Target object (Motorcycle) that violates a traffic signal from the left is driving straight.
4	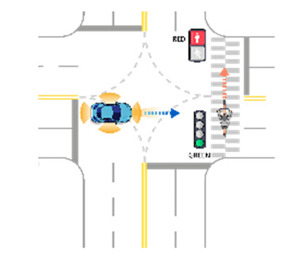	Driving straight	Jaywalking	Ego-vehicle is driving straight on the right of way at signalized intersection. Target object (Bicycle) is jaywalking on the other side.

**Table 8 sensors-21-06929-t008:** Ratio of vehicle-to-vehicle functional scenarios from real accident data on road sections.

No.	Provoking Events	Functional Scenarios	Ratios
1	Lane-change	Ego-vehicle is driving straight at road section. Target object (Vehicle) is lane-changing into ego-vehicle’s driving lane ahead.	22.70%
2	Abrupt lane-change	Ego-vehicle is driving straight at road section. Target object (Vehicle) is suddenly lane-changing into ego-vehicle’s driving lane ahead.	2.48%
3	Stopping	Ego-vehicle is driving straight at road section. Target object (Vehicle) is stopping in ego-vehicle’s driving lane ahead.	5.35%
4	U-turn	Ego-vehicle is driving straight at road section. Target object (Vehicle) is making a U-turn into ego-vehicle’s driving lane ahead.	4.25%
5	Abrupt stopping	Ego-vehicle is driving straight at road section. Target object (Vehicle) is suddenly stopping in ego-vehicle’s driving lane ahead.	2.37%
6	Driving over centerline	Ego-vehicle is driving straight at road section. Target object (Vehicle) is driving over centerline into ego-vehicle’s driving lane.	2.02%
7	Reversing	Ego-vehicle is driving straight at road section. Target object (Vehicle) is reversing into ego-vehicle’s driving lane.	0.18%

**Table 9 sensors-21-06929-t009:** Ratio of vehicle-to-pedestrian functional scenarios from real accident data on road sections.

No.	Provoking Events	Functional Scenarios	Ratios
1	Crossing	Ego-vehicle is driving straight at road section. Target object (Pedestrian) is crossing ahead.	1.74%
2	Walking	Ego-vehicle is driving straight at road section. Target object (Pedestrian) is walking in lane ahead.	0.04%
3	Jaywalking	Ego-vehicle is driving straight at road section. Target object (Pedestrian) is jaywalking ahead.	0.32%

**Table 10 sensors-21-06929-t010:** Ratio of vehicle-to-motorcycle functional scenarios from real accident data on road sections.

No.	Provoking Events	Functional Scenarios	Ratios
1	Lane-change	Ego-vehicle is driving straight at road section. Target object (Motorcycle) is lane-changing into ego-vehicle’s driving lane.	0.71%
2	U-turn	Ego-vehicle is driving straight at road section. Target object (Motorcycle) is making a U-turn into ego-vehicle’s driving lane.	0.18%
3	Stopping	Ego-vehicle is driving straight at road section. Target object (Motorcycle) is stopping in ego-vehicle’s driving lane ahead.	0.18%

**Table 11 sensors-21-06929-t011:** Ratio of vehicle-to-bicycle functional scenarios from real accident data on road sections.

No.	Provoking Events	Functional Scenarios	Ratios
1	Crossing	Ego-vehicle is driving straight at road section. Target object (Bicycle) is crossing ahead.	0.96%
2	Driving straight	Ego-vehicle is driving straight at road section. Target object (Bicycle) is driving straight into ego-vehicle’s lane ahead.	0.14%
3	Reversing	Ego-vehicle is driving straight at road section. Target object (Bicycle) is reversing into ego-vehicle’s driving lane.	0.07%

**Table 12 sensors-21-06929-t012:** Ratio of vehicle-to-vehicle functional scenarios from real accident data at intersection sections.

No.	Provoking Events	Functional Scenarios	Ratios
1	Violating traffic signal	Ego-vehicle is driving straight on the right of way at signalized intersection. Target object (Vehicle) that violates a traffic signal from the left is driving straight.	1.01%
2	Violating traffic signal	Ego-vehicle is driving straight on the right of way at signalized intersection. Target object (Vehicle) that violates a traffic signal from the right is driving straight.	1.01%
3	Violating traffic signal	Ego-vehicle is driving straight on the right of way at signalized intersection. Target object (Vehicle) that violates a traffic signal from the other side is turning right.	2.90%
4	Violating traffic signal	Ego-vehicle is driving straight on the right of way at signalized intersection. Target object (Vehicle) that violates a traffic signal from the other side is turning left.	1.03%
5	Lane-change	Ego-vehicle is driving straight on the right of way at signalized intersection. Target object (Vehicle) is changing lanes into ego-vehicle’s lane in the same direction.	1.90%
6	Stopping	Ego-vehicle is driving straight on the right of way at signalized intersection. Target object (Vehicle) is stopping in front of ego-vehicle in the same direction.	1.61%
7	Violating traffic signal	Ego-vehicle is turning left on the right of way at signalized intersection. Target object (Vehicle) that violates a traffic signal from the left is driving straight.	1.90%
8	Violating traffic signal	Ego-vehicle is turning left on the right of way at signalized intersection. Target object (Vehicle) that violates a traffic signal from the right is driving straight.	1.90%
9	Violating traffic signal	Ego-vehicle is turning left on the right of way at signalized intersection. Target object (Vehicle) that violates a traffic signal from the other side is turning left.	1.10%
10	Violating traffic signal	Ego-vehicle is turning left on the right of way at signalized intersection. Target object (Vehicle) that violates a traffic signal from the other side is turning right.	1.90%
11	Lane-change	Ego-vehicle is turning left on the right of way at signalized intersection. Target object (Vehicle) is changing lanes into ego-vehicle’s lane in the same direction.	0.31%
12	Stopping	Ego-vehicle is turning left on the right of way at signalized intersection. Target object (Vehicle) is stopping in front of ego-vehicle in the same direction.	3.22%
13	Violating traffic signal	Ego-vehicle is turning right on the right of way at signalized intersection. Target object (Vehicle) that violates a traffic signal from the other side is driving straight.	0.02%
14	Violating traffic signal	Ego-vehicle is turning right on the right of way at signalized intersection. Target object (Vehicle) that violates a traffic signal from the other side is turning left.	0.00%
15	Lane-change	Ego-vehicle is turning right on the right of way at signalized intersection. Target object (Vehicle) is changing lanes into ego-vehicle’s lane in the same direction.	0.00%
16	Stopping	Ego-vehicle is turning right on the right of way at signalized intersection. Target object (Vehicle) is stopping in front of ego-vehicle in the same direction.	0.00%

**Table 13 sensors-21-06929-t013:** Ratio of vehicle-to-pedestrian functional scenarios from real accident data at intersection sections.

No.	Provoking Events	Functional Scenarios	Ratios
1	Jaywalking	Ego-vehicle is driving straight on the right of way at signalized intersection. Target object (Pedestrian) is jaywalking ahead at crosswalk.	0.24%
2	Jaywalking	Ego-vehicle is turning left on the right of way at signalized intersection. Target object (Pedestrian) is jaywalking ahead at crosswalk.	0.24%
3	Jaywalking	Ego-vehicle is turning right on the right of way at signalized intersection. Target object (Pedestrian) is jaywalking ahead at crosswalk.	0.10%

**Table 14 sensors-21-06929-t014:** Ratio of vehicle-to-motorcycle functional scenarios from real accident data at intersection sections.

No.	Provoking Events	Functional Scenarios	Ratios
1	Violating traffic signal	Ego-vehicle is driving straight on the right of way at signalized intersection. Target object (Motorcycle) that violates a traffic signal from the left is driving straight.	0.91%
2	Violating traffic signal	Ego-vehicle is driving straight on the right of way at signalized intersection. Target object (Motorcycle) that violates a traffic signal from the right is driving straight.	0.91%
3	Violating traffic signal	Ego-vehicle is driving straight on the right of way at signalized intersection. Target object (Motorcycle) that violates a traffic signal from the other side is turning left.	1.92%
4	Violating traffic signal	Ego-vehicle is driving straight on the right of way at signalized intersection. Target object (Motorcycle) that violates a traffic signal from the other side is turning right.	1.06%
5	Lane-change	Ego-vehicle is driving straight on the right of way at signalized intersection. Target object (Motorcycle) is changing lanes into ego-vehicle’s lane in same direction.	0.17%
6	Stopping	Ego-vehicle is driving straight on the right of way at signalized intersection. Target object (Motorcycle) is stopping in front of ego-vehicle in the same direction.	0.14%
7	Violating traffic signal	Ego-vehicle is turning left on the right of way at signalized intersection. Target object (Motorcycle) that violates a traffic signal from the left is driving straight.	0.24%
8	Violating traffic signal	Ego-vehicle is turning left on the right of way at signalized intersection. Target object (Motorcycle) that violates a traffic signal from the right is driving straight.	0.24%
9	Violating traffic signal	Ego-vehicle is turning left on the right of way at signalized intersection. Target object (Motorcycle) that violates a traffic signal from the other side is turning left.	0.38%
10	Violating traffic signal	Ego-vehicle is turning left on the right of way at signalized intersection. Target object (Motorcycle) that violates a traffic signal from the other side is turning right.	0.02%
11	Lane-change	Ego-vehicle is turning left on the right of way at signalized intersection. Target object (Motorcycle) is changing lanes into ego-vehicle’s lane in the same direction.	0.07%
12	Stopping	Ego-vehicle is turning left on the right of way at signalized intersection. Target object (Motorcycle) is stopping in front of ego-vehicle in the same direction.	0.22%
13	Violating traffic signal	Ego-vehicle is turning right on the right of way at signalized intersection. Target object (Motorcycle) that violates a traffic signal from the other side is driving straight.	0.17%
14	Violating traffic signal	Ego-vehicle is turning right on the right of way at signalized intersection. Target object (Motorcycle) that violates a traffic signal from the other side is turning left.	0.10%
15	Lane-change	Ego-vehicle is turning right on the right of way at signalized intersection. Target object (Motorcycle) is changing lanes into ego-vehicle’s lane in the same direction.	0.05%
16	Stopping	Ego-vehicle is turning right on the right of way at signalized intersection. Target object (Motorcycle) is stopping in front of ego-vehicle in the same direction.	0.10%

**Table 15 sensors-21-06929-t015:** Ratio of vehicle-to-bicycle functional scenarios from real accident data at intersection sections.

No.	Provoking Events	Functional Scenarios	Ratios
1	Jaywalking	Ego-vehicle is driving straight on the right of way at signalized intersection. Target object (Bicycle) is jaywalking on other side.	0.22%
2	Jaywalking	Ego-vehicle is turning left on the right of way at signalized intersection. Target object (Bicycle) is jaywalking on other side.	0.12%
3	Jaywalking	Ego-vehicle is turning right on the right of way at signalized intersection. Target object (Bicycle) is jaywalking on other side.	0.22%

## Data Availability

The data used in this study cannot be made available due to the policy of the Korean National Police Agency (KNPA).

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
