# Peer review of "Scenario-Mining for Level 4 Automated Vehicle Safety Assessment from Real Accident Situations in Urban Areas Using a Natural Language Process"

_sensors, 2021, doi:10.3390/s21206929_

Round 1
Reviewer 1 Report
The approach to use Natural language processing for traffic accident analysis or classification is a promising approach and the motivation for the paper is timely and important. The authors clearly defined the ODD, which is a clever "move" not to solve every problem, though the restricted search area is eligible for the proof of concept.
Though the intention and the toolchain are clear for the prepared reader, more details and insight should be given in the text.
My concerns:
- Though it is more philosophical than practical, is it a good approach to use "human-generated" accidents to provide scenarios for AVs? Is it possible that AVs fail in different situations or under different conditions? Also, Lvl4 as a target is not enough, each AV solution needs its own circumstances like communication fails, the sensor fails, etc... These need more detail in the paper.
- The literature is shallow and provides an unprofessional look to the paper. Much more background is needed for the NLP, its uses for close or similar domains, and also for scenario generation, which has an enormous literature in the past few years. Also, the discussion of the strengths and weaknesses of other approaches should be appreciated.
- Section 3.1 (especially the table) is omittable, as SAE levels are part of the common knowledge in this area.
- The representation of methodology needs more detail.
- It is a matter of style, though I would keep a few examples of the scenarios in the text, and enumerate them in the appendix.
- What are the "the meaningless features"? This should be detailed, which features were neglected or highlighted.
- The paper should discuss if there are any scenarios that have been left out?
Author Response
Thank you for your comments and suggestions. We have thoroughly reviewed our manuscript based on your comments and suggestions, and finally made edits, corrections, and revisions throughout the entire manuscript. Please look into our revised manuscript, and feel free to let us know if you have further comments. We would appreciate that.

Reviewer 2 Report
The authors proposed a test scenario mining methodology from traffic accident data using NLP for automated vehicle safety assessment. Test scenario generation is an essential part of the automated vehicles verification and validation process.
There seem to be 3 section 5s in this paper, i.e., "5. Results", "5. Scenario Development Results", and "5. Validation". Therefore, it's recommended to make the latter two sub-sections of section "5. Results". Besides that, the paper overall has a good structure with seven sections: "1. Introduction ", "2. Literature Review", "3. Premise of Scenario Development", "4. Methodology", "5. Results", "6. Conclusions", and "7. Recommendations for Future Research". Also, it's suggested for authors to perform a more extensive literature review to better support this paper.
Verification and validation terms have different definitions and are not the same; the reviewer suggests changing the section "5. Validation" title to reflect that section better. Validation of this methodology would require actual testing and experiments, for instance, offline testing using simulation, which authors have noted as a recommendation for future research.
This methodology seems to be heavily based on the structure of the Korean traffic accident dataset KNPA. The reviewer suggests further discussing and noting if this method can be generalized to other traffic accident datasets from different countries and how it can be used.
Multiple paragraphs throughout the paper need rewording to convey the authors' message better. Proofreading and further editing are advised.
Author Response

(The authors gave the same response as above.)

Round 2
Reviewer 1 Report
Thanks for the reply.